# Biological Characterization of Human Autologous Pericardium Treated with the Ozaki Procedure for Aortic Valve Reconstruction

**DOI:** 10.3390/jcm10173954

**Published:** 2021-08-31

**Authors:** Chiara Gardin, Giampaolo Morciano, Letizia Ferroni, Elisa Mikus, Alberto Tripodi, Maurizio Pin, Elena Tremoli, Alberto Albertini, Barbara Zavan

**Affiliations:** 1GVM Care & Research, Maria Cecilia Hospital, 48033 Cotignola, Italy; cgardin@gmnet.it (C.G.); lferroni@gvmnet.it (L.F.); emikus@gmnet.it (E.M.); albertotripodi@hotmail.com (A.T.); maurizio.pin10@gmail.com (M.P.); etremoli@gmnet.it (E.T.); aalbertini@gmnet.it (A.A.); 2Section of Experimental Medicine, Laboratory for Technologies of Advanced Therapies (LTTA), Department of Medical Sciences, University of Ferrara, 44121 Ferrara, Italy; giampaolo.morciano@unfe.it; 3Department of Translational Medicine, University of Ferrara, 44121 Ferrara, Italy

**Keywords:** aortic valve reconstruction, human autologous pericardium, Ozaki procedure, cardiac surgery, glutaraldehyde treatment, re-endothelialization

## Abstract

Background: The Ozaki procedure is an innovative surgical technique aiming at reconstructing aortic valves with human autologous pericardium. Even if this procedure is widely used, a comprehensive biological characterization of the glutaraldehyde (GA)-fixed pericardial tissue is still missing. Methods: Morphological analysis was performed to assess the general organization of pericardium subjected to the Ozaki procedure (post-Ozaki) in comparison to native tissue (pre-Ozaki). The effect of GA treatment on cell viability and nuclear morphology was then investigated in whole biopsies and a cytotoxicity assay was executed to assess the biocompatibility of pericardium. Finally, human umbilical vein endothelial cells were seeded on post-Ozaki samples to evaluate the influence of GA in modulating the endothelialization ability in vitro and the production of pro-inflammatory mediators. Results: The Ozaki procedure alters the arrangement of collagen and elastic fibers in the extracellular matrix and results in a significant reduction in cell viability compared to native tissue. GA treatment, however, is not cytotoxic to murine fibroblasts as compared to a commercially available bovine pericardium membrane. In addition, in in vitro experiments of endothelial cell adhesion, no difference in the inflammatory mediators with respect to the commercial patch was found. Conclusions: The Ozaki procedure, despite alteration of ECM organization and cell devitalization, allows for the establishment of a noncytotoxic environment in which endothelial cell repopulation occurs.

## 1. Introduction

Human autologous pericardium has been used for the reconstruction of aortic valve since the late 1960s [1,2]. These first attempts with untreated autologous pericardial tissue have been generally unsatisfactory due to tissue thickening and shrinkage, resulting in structural and functional valve degeneration with the need for reoperation. In order to limit the tissue retraction problems, some years later Love and colleagues first suggested the immersion of pericardium in a solution of 0.6% glutaraldehyde (GA) [3]. In 1995, the idea was taken up by Duran and co-workers [4], then refined by Ozaki et al. more recently [5].

From a methodological point of view, the Ozaki procedure involves the treatment of a portion of human autologous pericardium with a solution of 0.6% GA for 10 min, followed by three serial washings of 6 min each in a saline solution. Treatment with GA has been shown to facilitate the handling of pericardium, by providing greater resistance to retraction, while maintaining the intrinsic flexibility of the tissue, and by stabilizing its shape, thanks to the crosslinking of tissue proteins [6]. In addition to improving its mechanical properties, treatment with GA reduces the antigenicity of the pericardium, its thrombogenicity and increases its resistance to enzymatic degradation [7]. On the other hand, the use of GA has been implicated in the process of calcification, which is in turn correlated to the long-term dysfunction of bioprostheses. The mechanism of calcification is considered a multi-factorial event, mainly triggered by the interaction between membrane phospholipids of cells devitalized, but not removed by the GA treatment, and the increased influx of extracellular calcium, which leads to the formation of hydroxyapatite [8,9]. To some extent, the extracellular matrix (ECM) structural proteins collagen and elastin also represent nucleation sites for calcium deposition [10].

The main difference between the Ozaki procedure and the others is represented by the independent replacement of each valvular cusp by autologous pericardial cusps reflecting the size of the natural valve leaflets. The size of each aortic cusp is measured based on the distance between the commissures [5]. From 2007 to 2019, this technique has been employed with success on more than 1100 patients affected by a variety of aortic valve diseases (e.g., aortic stenosis, aortic regurgitation, infective endocarditis, prosthetic valve endocarditis, annuloaortic ectasia), and with all types of aortic valve anatomies [11,12,13]. Actually, mid-term follow-up studies (7 years after the procedure) revealed no signs of calcification nor aortic regurgitation, and natural motion of the three leaflets [11].

Despite the increasing use of the Ozaki surgical procedure, a comprehensive evaluation of the biological properties of GA-treated pericardial tissue is still missing to date. In light of these considerations, the aim of the present study was the characterization from a biological point of view of human autologous pericardium used for the reconstruction of aortic valve according to the Ozaki procedure. In this study, histological examination made it possible to evaluate the organization of pericardium before and after GA treatment, whereas scanning electron microscopy (SEM) was performed to observe the appearance of the pericardium surfaces under these two conditions. We then investigated the effect of the GA treatment on cell viability and nuclear morphology in whole biopsies; next, a cytotoxicity assay was carried out to assess the biocompatibility of pericardium in vitro. Finally, the ability of human endothelial cells to repopulate in vitro the GA-treated pericardium and their biological performance was investigated in terms of cell adhesion and pro-inflammatory marker release.

## 2. Materials and Methods

### 2.1. Sample Collection and Processing

Human autologous pericardium was obtained as discharge material from patients (*n* = 15) undergoing aortic valve reconstruction during heart surgery. All the patients had provided written and informed consent to sample collection and all other procedures described herein. The protocol was approved by the Ethics Committee of Maria Cecilia Hospital (PARACADUTE study, version 1, 15 March 2021). Human pericardium samples (7 × 8 cm) were separated from surrounding tissues; then, a native 2 × 2 cm sample was removed and used as control material for comparison purposes (pre-Ozaki samples), whereas the remaining tissue was treated with GA according to the literature [5]. The fixed tissue was used to create the neo-leaflets, with the rest of the material composing the post-Ozaki samples. Human pericardia were then transferred to the laboratory and stored refrigerated until further processing.

### 2.2. Histological Analyisis

For histological examinations, the post-Ozaki and pre-Ozaki samples (0.5 × 0.5 cm) were fixed in 4% paraformaldehyde, embedded in paraffin, then cut into 7 µm thick sections. For Hematoxylin/Eosin (H/E) staining, pericardium sections were stained with the nuclear dye hematoxylin (Kaltek S.r.l., Padova, Italy) for 10 min and the counterstain eosin (Kaltek S.r.l.) for 1 min. For the visualization of collagen and elastic fibers, pericardium sections were stained with Weigert’s solution (Sigma-Aldrich, St. Louis, MA, USA) for 10 min, differentiated in a solution of ferric chloride (Sigma-Aldrich), then stained in van Gieson’s solution (Sigma-Aldrich) for 1 min. Images were taken with a Nikon Eclipse Ni upright microscope (Nikon Corporation, Tokyo, Japan) equipped with a 20× objective, and analyzed with the Fiji software VS5 [14].

### 2.3. SEM

Tissue biopsies (0.5 × 0.5 cm) were fixed in 2% GA prepared in 0.1 M 4-(2-hydroxyethyl)-1-piperazineethanesulfonic acid (HEPES) buffer O/N at +4 °C. Samples were then washed twice with 0.1 M HEPES buffer and progressively dehydrated by subsequent immersion in increasing concentrations of ethanol for 15 min each. Before SEM imaging, the samples were subjected to chemical drying with hexamethyldisilazane (Sigma-Aldrich), which was finally removed by complete evaporation. All micrographs were obtained using a Zeiss EVO 40 SEM microscope (Zeiss, Jena, Germany) at Centro di Microscopia Elettronica (University of Ferrara, Ferrara, Italy).

### 2.4. MTT Assay

To determine cell viability in the post-Ozaki and pre-Ozaki biopsies, the methyl-thiazolyl-tetrazolium (MTT)-based proliferation assay was performed, as published in Gardin et al. with minor modifications [15]. Briefly, tissue samples (0.5 × 0.5 cm) were incubated at 37 °C for 3 h in 0.6 mL of 0.5 mg/mL MTT solution prepared in phosphate buffered saline (PBS, EuroClone, Milan, Italy). After removal of the MTT solution, 0.6 mL of 10% dimethyl sulfoxide in isopropanol was added to extract the formazan in the samples for 60 min at 37 °C. For each sample, optical density values at 570 nm were recorded in duplicate on aliquots deposited in 96-well culture plates using a multilabel plate reader (Multiskan™ FC, Thermo Fisher Scientific, Waltham, MA, USA). Cell viability in post-Ozaki samples was expressed as a percentage with respect to pre-Ozaki samples (considered as 100%).

### 2.5. Live/Dead Assay

To distinguish live and dead cells in the post-Ozaki and pre-Ozaki samples, the fluorescent LIVE/DEAD^®^ Viability/Cytotoxicity Kit for mammalian cells (Thermo Fisher Scientific) was used. In detail, 2 µM Calcein-AM and 4 µM Ethidium homodimer-1 were used to stain live cells (green) and dead cells (red), respectively, during 1 h incubation of tissue samples (0.5 × 0.5 cm) at 37 °C. A laser scanning confocal microscopy system (Nikon A1 confocal microscope, Nikon Corporation, Tokyo, Japan) equipped with a 63× objective was used for image acquisition. Cell viability was measured with the Cell Counter tool of Fiji software [14]. Live cells were quantified by dividing the green stained cells by total number of cells, while dead cells were calculated by dividing the red stained cells by the total number of cells. For each sample, eight random fields were counted. Cell viability in post-Ozaki samples was then expressed as a percentage with respect to pre-Ozaki samples (considered as 100%). Cell death in post-Ozaki has been expressed as fold change with respect to pre-Ozaki samples.

### 2.6. Nuclear Morpholgy Analysis

The post-Ozaki and pre-Ozaki tissue samples (0.5 × 0.5 cm) were incubated with 2 µg/mL of the Hoechst 33342 nuclear probe (Thermo Fisher Scientific) for 60 min at 37 °C, then observed under the Nikon A1 confocal microscope at 63× magnification. The acquired images were analyzed with the Fiji software for measuring intensity of nuclear staining, nuclear circularity factor and size of the nuclei [14]. For each sample, at least ten nuclei were analyzed.

### 2.7. Cytotoxicity Assay

The in vitro cytotoxic potential of GA-treated human pericardium was evaluated by a direct contact method following ISO 10993-5 standards [16]. Test samples (post-Ozaki samples), negative control (commercially available bovine pericardial patch, BP, C0914, St. Jude Medical, Saint Paul, MN, USA), and positive control (cells cultured in the presence of 0.6% GA) in triplicate were placed on a subconfluent monolayer of NCTC clone L929 mouse fibroblast cells (ATCC, Manassas, VA, USA). Samples covered approximatively one tenth of the cell layer surface. Cells cultured on tissue culture polystyrene without tested materials represented the blank condition. After incubation of cells under the different conditions at 37 ± 2 °C for 24 ± 1 h, the cytotoxicity of the materials was examined both microscopically for cellular response around the samples and by quantifying the cell metabolic activity with the MTT assay. The data were compared with the blank sample.

### 2.8. Cell Seeding onto Pericardium Samples and Biomarkers Analysis

Human umbilical vein endothelial cells (HUVEC, Thermo Fisher Scientific cat. C0155C) were cultured in complete medium 200 without phenol red (cM200PRF), which was made of medium 200 (Thermo Fisher Scientific cat. M200PRF500) supplemented with Low Serum Growth Supplement Kit (Thermo Fisher Scientific cat. S003K). At 80% confluence, HUVEC were detached with trypsin/EDTA (Thermo Fisher Scientific) and labelled with 2 µM Calcein-AM and 2 µg/mL Hoechst 33342 for 20 min at 37 °C. After labelling, HUVEC were seeded onto the fibrous surface of the post-Ozaki samples (0.5 × 0.5 cm) by gently pipetting 50 µL of the cell suspension containing 5 × 10^5^ of cells. Cells seeded on the rough surface of a commercial BP patch (C0914, St. Jude Medical) represented the comparative group. One hour after seeding, 0.5 mL of cM200PRF was added to the wells and cells were cultured for 4 days. Cells were then imaged under the Nikon A1 confocal microscope at 40× magnification. The number of cells adhering on pericardial samples was measured with the Cell Counter tool of Fiji software [14] and expressed as number of cells per field (0.09 mm^2^). For each sample, eight random fields were analyzed.

Culture media collected from each scaffold after 4 days were assayed for the production of high mobility group box 1 (HMGB1) protein and interleukin-6 (IL-6) using commercial sandwich ELISA kits according to the manufacturer’s instructions. Biomarker concentrations was determined by comparing data to standards obtained with test kits.

### 2.9. Statistical Analysis

All data sets were tested for normal distribution using the Shapiro–Wilk normality test. For all data sets that were normally distributed, the statistical methods included Student’s *t*-test for comparative analysis between two groups, and the one-way ANOVA test for comparative analysis of three or more groups using the GraphPad 8.0 software. Data were expressed as mean ± standard error of the mean (SE), and statistical significance was set at *p* < 0.05. For other not normal distributions, the non-parametric Mann–Whitney U test was used. Data were expressed as median ± interquartile range and statistical significance was set at *p* < 0.05. All experiments were conducted in triplicate.

## 3. Results

### 3.1. The Ozaki Procedure Alters the Arrangement of Collagen and Elastic Fibers in the ECM

The effect of the GA treatment on the general organization of human pericardium was investigated by means of morphological analysis on tissue biopsies. The histological appearance of pericardial samples before and after the Ozaki procedure is illustrated in Figure 1A–D. H/E staining identified the presence, in both the pre- and post-Ozaki biopsies, of the serosa component of pericardium consisting of a single layer of mesothelial cells. This layer, however, was continuous in the native tissue (Figure 1A), whereas it was interrupted in the pericardium after the Ozaki procedure (Figure 1B). Subjacent to the mesothelium, the fibrosa was composed of dense collagen bundles with interspersed scant elastic fibers. In this layer, fibroblasts nuclei and blood vessels were also visible.

Weigert van Gieson staining was used to qualitatively assess the content and distribution of the collagen bundles and elastic fibers in the ECM. The GA treatment was not found to cause severe tissue alteration, as witnessed by an overall well-preserved ECM with collagen bundles (Figure 1C,D, in red) and interspersed short elastic fibers (Figure 1C,D, in black). Compared to controls, post-Ozaki tissues were denser and compact with reduced spacing between the collagen bundles. In addition, both the fibers lost their wavy orientation in favor of a more stretched one.

SEM analysis was then carried out to visualize surface characteristics of human autologous pericardium subjected to the Ozaki procedure. The surface of the serosa side of pre- and post-Ozaki samples is shown in Figure 2A–D. In the native pericardium, the serous layer has a smooth appearance consisting of a continuous layer of mesothelial cells (Figure 2A), flat and cuboidal epithelial cells rich in microvilli (Figure 2C), which are essential for the secretion and reabsorption of pericardial fluid. On the contrary, SEM micrographs of the GA-treated pericardium showed areas in which mesothelial cells have been denuded (Figure 2B), thus exposing the underlying fibrous layer (Figure 2D, area delimited by the red box). SEM images obtained from the fibrous side of the pre-Ozaki pericardium evidenced the presence of irregularly dispersed and wavy collagen bundles and elastic fibers (Figure 2E,G), in contrast to the pericardium treated by the Ozaki procedure, where a more parallel arrangement of the collagen and elastic fibers in pericardium was observable (Figure 2F,H).

### 3.2. The Ozaki Procedure Triggers Cell Death in Pericardium

To evaluate the biological properties of the whole tissue samples treated by the Ozaki procedure, we first performed a cell viability assay. We selected the MTT protocol to compare the residual cell metabolic activity of post-Ozaki compared to native pericardium. The Ozaki procedure determined a significant (*p* < 0.0001) reduction in pericardial cell viability with respect to pre-Ozaki tissues (Figure 3A). In particular, the MTT assay on GA-treated pericardium showed 33.32 ± 1.29% of metabolic activity compared to that of native pericardium (100.00 ± 3.85%).

With the same purpose, and taking advantage of confocal microscopy techniques, we analyzed the most superficial layers of both experimental conditions by either quantifying the signal of the Calcein-AM probe, a marker of cell viability, or ethidium bromide (EtBr), a marker of cell death, in pre-Ozaki (Figure 3B) and post-Ozaki (Figure 3C) samples. We detected a strong reduction in viability of cells located in the most superficial layers in favor of cell death due to GA treatment; in fact, when compared to native pericardium (median: 87.7%; 25% percentile: 13.70%; 75% percentile: 187.90%), the GA-fixed tissue showed 0% median value (25% percentile: 0%; 75% percentile: 2.74%) of live cells (*p* < 0.0001) (graph in Figure 3B). Conversely, the percentages of dead cells were threefold higher in post-Ozaki compared to pre-Ozaki samples (graph in Figure 3C).

Analysis of nuclear morphology is used as an additional indicator of the state of cell health [17]. For this reason, we investigated whether nuclear morphology was also affected by GA treatment in terms of nuclei dimension, circularity, and intensity measured by Hoechst probe staining. First, we detected a different spatial rearrangement of cells from post-Ozaki samples that resulted to be mostly located on a single-like layer, rather than organized in three dimensions, as for pre-Ozaki. Second, GA produced a significant (*p* < 0.01) number of jagged nuclei as evidenced by analyzing the circularity factor, which was 0.49 ± 0.04, in contrast to the nuclear circularity in the native pericardium of 0.85 ± 0.07 (Figure 3D, center). In addition, a significantly (*p* < 0.05) higher Hoechst intensity, which is an indication of an increase in cell membrane permeability, was found in the GA-treated pericardium (1.50 ± 0.13) when compared to untreated tissues (1.01 ± 0.08) (Figure 3D, right). Nuclei dimension was not affected by the chemical crosslinking (98.4 ± 6.94 and 76.48 ± 9.00 in the pre- and post-Ozaki samples, respectively; Figure 3D, left).

### 3.3. The Ozaki Procedure Is Not Toxic and Allows for Endothelial Cell Repopulation In Vitro

We next examined the in vitro cytotoxicity of the GA-treated pericardium, which is an essential parameter to consider when a material has to be implanted in the human body, directly in contact with living tissues. The commercial BP patch, as well as post-Ozaki samples, were found to be noncytotoxic to L929 fibroblasts by a direct contact method, as revealed by cell morphologies (Figure 4C,D, respectively), similar to that of the blank sample (Figure 4A). By quantitatively measuring cytotoxicity, the MTT assay on L929 cells in contact with the GA-treated pericardium and negative control showed 87.51 ± 1.20% and 86.87 ± 2.32% of cell metabolic activities, respectively, in relation to the blank (100.0 ± 1.75%), without significant differences between the two conditions (Figure 4E). In contrast, the positive control was severely cytotoxic towards L929 cells, as demonstrated by both a compromised cell morphology (Figure 4B) and a measured metabolic activity of 2.53 ± 0.03% (Figure 4E) when compared to that of the blank control.

In order to evaluate the endothelialization ability of human pericardium after the Ozaki procedure, endothelial cells were labelled with the Calcein-AM and Hoechst probes, then seeded on the fibrous layer of both post-Ozaki tissues and BP control membranes. After 4 days, cells positive to the probes were counted with confocal microscopy. Figure 4F shows that the number of HUVEC grown onto post-Ozaki samples were 9.72 ± 0.52 per field (0.09 mm^2^), whereas there were 6.59 ± 0.48 cells on the BP control substrate with the same area (*p* < 0.0001). Fluorescent images of cytoplasmic and nuclei staining of the cells incubated for 4 days on the pericardial substrates revealed that these were attached and distributed on the scaffold, exhibiting their characteristic morphology (Figure 4F).

Finally, we investigated whether endothelial cells culturing on the GA-treated pericardium may impact the expression of some pro-inflammatory mediators. In particular, we quantified HMGB1 and IL-6 levels in the supernatant harvested 4 days after HUVEC seeding on post-Ozaki samples and BP membranes. As shown in Figure 4G,H, significantly higher levels of both inflammatory molecules were released from HUVEC cultured on the GA-fixed pericardium than control.

## 4. Discussion

In recent years, the Ozaki procedure has attracted increasing interest due to the potential numerous advantages of GA treatment on human autologous pericardium. Upon reviewing the literature, however, a complete evaluation of the biological effect of GA on human pericardium is still missing. Therefore, this study was designed to provide a comprehensive in vitro biological characterization of human autologous pericardium treated by the Ozaki procedure for the reconstruction of the aortic valve.

To this end, histological analysis of pericardial biopsies was performed in order to investigate morphological changes that occur in the pericardium in response to treatment with GA. H/E staining was used to determine the general morphology of the tissue and the degree of the tissue histo-architecture preservation after GA crosslinking. Compared to the native pericardium, post-Ozaki samples presented good maintenance of the ECM structure. The most remarkable alterations caused by the Ozaki procedure concerned the waviness of collagen and elastic fibers. In fact, as evidenced by the Weigert van Gieson staining, post-Ozaki pericardial samples exhibited densely packed parallel bundles of collagen and elastic fibers. These observations were confirmed by SEM analysis of the fibrosa layer of post-Ozaki samples, showing that both of these fibers assumed a more parallel arrangement. This effect may depend upon the procedure itself, which involves chemical fixation of the pericardium under stretch onto appropriate plates. As a result, the wavy structure of the fibers is lost during stretching and the simultaneous GA treatment, by crosslinking collagen molecules, maintains the stretched arrangement even after the removal of the stretch. From a mechanical point of view, the implication of such an elongation might be related to an increase in tensile strength observed in the GA-treated autologous pericardium with respect to native aortic valve leaflets [18]. Both the histological and SEM analysis also revealed that the mesothelial side of the GA-treated pericardium displayed denudation across the surface, which may be due to tissue processing, as described by others for bovine pericardium [19].

We also assessed the effect of the GA treatment on the cellular component of autologous pericardium by analyzing cell viability and nuclear morphology in whole biopsies. Therefore, the residual cell viability in post-Ozaki samples was investigated by means of the MTT test and live/dead fluorescence assay and compared to that of untreated tissues. Both assays showed that post-Ozaki samples had a significantly lower cell viability than native tissues. This is not surprising, since GA treatment is known to be responsible for the devitalization of cells [9]. Nonetheless, the live/dead assay evidenced a more pronounced reduction in cell viability of the GA-treated pericardium when compared to the MTT assay. The different indices of cell viability between the MTT assay and the live/dead assay may be explained by both the different sensitivity levels of the two techniques and the diverse nature of the used methodologies. The absorbance-based detection method used by the MTT assay is generally less sensitive than fluorescent methods for detecting the number of viable cells [20]. In addition, while the MTT assay measures the amount of living cells based on how many functional mitochondria are present, thus reflecting cell metabolism, the live/dead assay gives indication of cell viability based on both the intracellular esterase activity and cell membrane integrity. On the other hand, immunofluorescence analysis only allows the observation of the most superficial layers of a biopsy and does not allow the obtaining of information on the cells located deeper and embedded in the ECM, as are the fibroblasts of the fibrous layer of pericardium. Overall, these data indicate that in vitro cell viability should be evaluated by more than one test, in order to avoid over- or underestimation of the effect of experimental treatments, as suggested by others [21]. Alteration of nuclear morphology is often used as an indicator of dying cells, and it is assessed by nuclear staining using cell-permeable fluorescent dyes [17]. Healthy cells display diffuse staining of the nucleus, whereas injured or dead cells present DNA fragmentation and nuclear condensation. In this study, treatment of autologous pericardium with GA caused condensation of the nuclei, which were also stained more intensely with the Hoechst probe. It is reported that an increase in Hoechst intensity is strongly correlated to an increase in cell membrane permeability [22]. Altogether, these data not only further corroborate the devitalizing effect of GA, but also show that such a chemical crosslinking causes cell death by affecting both the biochemical and physical properties of the cells of the pericardium.

One of the problems generally associated with the use of GA-treated pericardium is the lack of bioactive properties of the tissue, which may result in valve calcification and subsequent failure over time [23]. It has been reported that toxic effects of the residual GA are mainly responsible for this phenomenon [7,24]. With the aim of testing whether GA treatment leaves cytotoxic residues that may prevent further host cells adhesion, migration and survival after tissue implantation, we first investigated the cytotoxicity of the GA-treated tissues; then, we tested their ability to be repopulated in vitro.

Cytotoxicity assay measures whether a material contains biologically harmful substances, thus providing evidence of the material’s biocompatibility [7]. The direct contact method was chosen, since it better mimics the physiological condition and allows evaluation of the diffusion zone, i.e., whether there is a toxic chemical concentration gradient around the material [25]. Our data indicate that human pericardium treated by the Ozaki procedure is not cytotoxic to murine fibroblasts, as revealed by the MTT quantification of cell metabolic activity, which was found to be around 87% when compared to that of the control cell cultures. According to the ISO 10993-5 standard, a material is considered cytotoxic when there is a reduction in cell viability of more than 30% [26]. Interestingly, the post-Ozaki samples behaved very similarly to the bovine pericardium used as negative control. This commercial membrane, which is subjected to a treatment with GA analogous to that of human pericardium, is routinely used for a wide range of cardiac and vascular repairs, and it is also well suited for aortic valve replacement [27]. The MTT data were confirmed by the qualitative microscopical visualization of the cells grown in contact with the tested materials, whose morphology appeared normal also in areas around or under the samples.

Cellular repopulation describes the ability of the tissue to be revitalized with the patient’s cells after implantation, and represents one of the methods for establishing the recovery of the biological properties of the GA-fixed tissue [28]. After having ascertained the absence of toxic effects of the residual GA in the treated autologous pericardium, in the last part of our work we wanted to assess whether the physiological endothelialization of the post-Ozaki samples was affected by the chemical treatment. To answer this question, endothelial cells were seeded on the fibrous layer of the GA-treated pericardium, which in the native valve is the most proximal layer contacting blood [29]. Then, the number of cells was compared to that of HUVEC adhering on the BP control membranes after 4 days of culture, as performed by others on bovine pericardium substrates [30,31]. Endothelial cells have been chosen because, among the different cell populations, they better mimic a physiological environment in cardiovascular applications [19]. In particular, endothelial cells play a regulatory role to the cardiac valve function, by sensing and integrating biological and mechanical signals from the blood, and by transmitting them to the underlying tissue [24]. Therefore, endothelialization of scaffolds implanted in cardiovascular sites improves their mechanical strength and limits the risk of inflammatory complications, thus reducing incidence of thrombosis and calcification [32,33]. Our data show that a greater number of cells adhered on the fibrous layer of the GA-treated pericardium with respect to the commercial bovine membrane. In addition, after staining with the Calcein-AM and Hoechst probes, it was observed that these cells were located on multiple focal planes, indicating that the topography of pericardium fibrous layer enabled the cells to penetrate into the tissue.

Finally, the production of inflammatory mediators by HUVEC seeded on post-Ozaki pericardium was investigated in order to evaluate whether GA affects endothelial cell function. HMBG1 is a non-histone chromatin-associated protein, which in basal conditions is found in the nucleus, where it favors the interaction of some transcription factors with DNA. When this protein is secreted in the extracellular space under inflammatory conditions, it acts as a pro-inflammatory cytokine [34]. It has been reported that HMGB1 is released also by endothelial cells subjected to an inflammatory stimulus, and functions as a mediator of cell migration [35,36]. IL-6, on the other hand, is a multifunctional cytokine produced by different cell types, including endothelial cells [37]. IL-6 has been reported to drive endothelial cell proliferation and migration in a dose-dependent manner, in association with enhancement of VEGF synthesis and release [38,39,40,41,42]. In this study, we found significant differences between the production of HMGB1 and IL-6 from the endothelial cells seeded onto post-Ozaki samples and those cultured onto BP control membranes, with both molecules being released in greater amounts from the GA-treated substrates. Nonetheless, it is also true that a significantly greater number of repopulating cells adhered on the GA-treated biopsies. Therefore, the increased amount of secreted proteins by post-Ozaki tissue might be due to the greater cell number that adhered to this tissue. Indeed, when levels of HMGB1 or IL-6 were normalized for cells number, no difference in the secretion of either biomarker was found between the two surfaces. This suggests that treatment of human autologous pericardium with GA does not evoke an inflammatory phenotype in endothelial cells. Further studies, however, are still needed to assess the effect of increasing time of endothelialization on the secretion of these biomarkers. In addition, it would be interesting to examine whether GA treatment impacts other functions of the repopulating endothelial cells, and whether pericardium endothelialization might be affected upon exposure to fluid shear stress, representative of blood flow, thus providing new insights into the hemodynamic of the pericardial tissue in vitro. Understanding whether stable and functional endothelial cell layers can be formed successfully on GA-fixed pericardium could serve as an indication of the durability of the aortic valve reconstructed with the Ozaki procedure.

## 5. Conclusions

To date, there has been limited characterization of human autologous pericardium treated with GA and used for the reconstruction of aortic valve. In this study, we show that the Ozaki surgical procedure alters the arrangement of collagen and elastic fibers in the ECM, which tend to assume a more stretched and compact orientation. In addition, treatment with GA triggers cell death in pericardium, compromising cell metabolic activity, plasma membrane permeability and nuclear organization. Nonetheless, the residual GA is not cytotoxic to murine fibroblasts and allows for endothelial cell repopulation in vitro, with no differences in the secretion of some inflammatory mediators when compared to a commercial bovine pericardium membrane, at least according to the experimental conditions explored in this study.

Taken together, these findings provide evidence that the Ozaki surgical procedure, despite alteration of pericardium ECM organization and cell devitalization, allows for the establishment of a noncytotoxic environment in which endothelialization occurs.

## Figures and Tables

**Figure 1 jcm-10-03954-f001:**
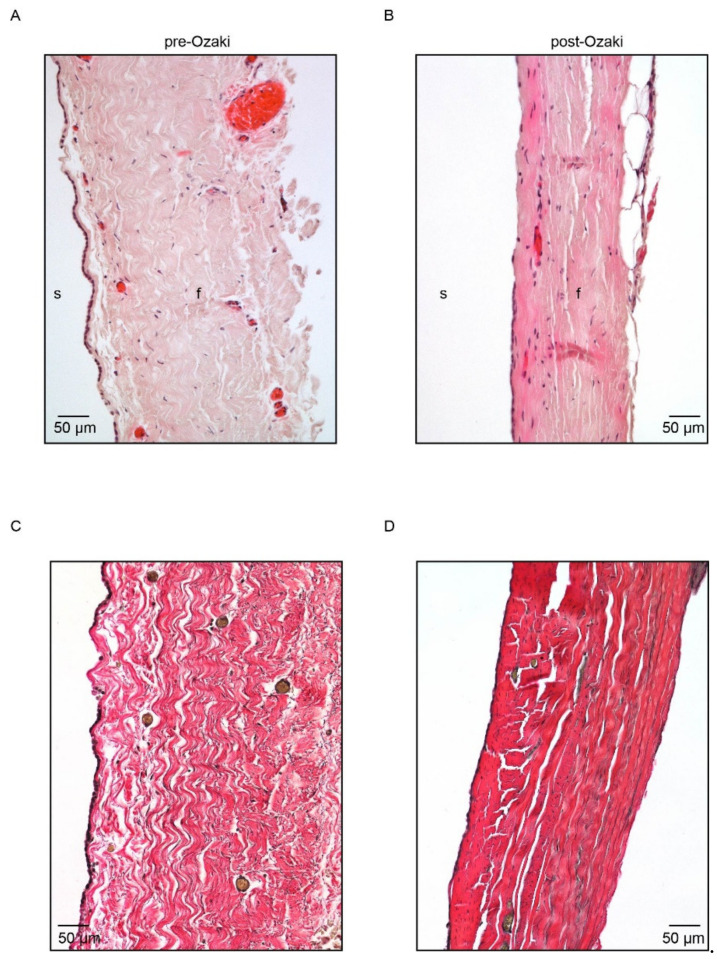
Histological characterization of pre-Ozaki and post-Ozaki samples. (**A**) Representative H/E image of the pre-Ozaki pericardium showing an intact layer of serosa (s) made up of mesothelial cells and a thicker fibrosa (f) layer consisting of collagen bundles and elastic fibers produced by its constituent fibroblasts. (**B**) H/E staining of the post-Ozaki pericardium shows preservation of the native ECM; nevertheless, the serous layer appears interrupted, whereas the fibrous layer is denser and compact. (**C**) Representative image of native pericardium after Weigert van Gieson staining shows variously oriented and wave-like collagen fibers (in red) with interspersed small elastic fibers (in black). (**D**) Weigert van Gieson staining of GA-treated pericardium shows changes in the architectures of collagen and elastin, which appear more stretched.

**Figure 2 jcm-10-03954-f002:**
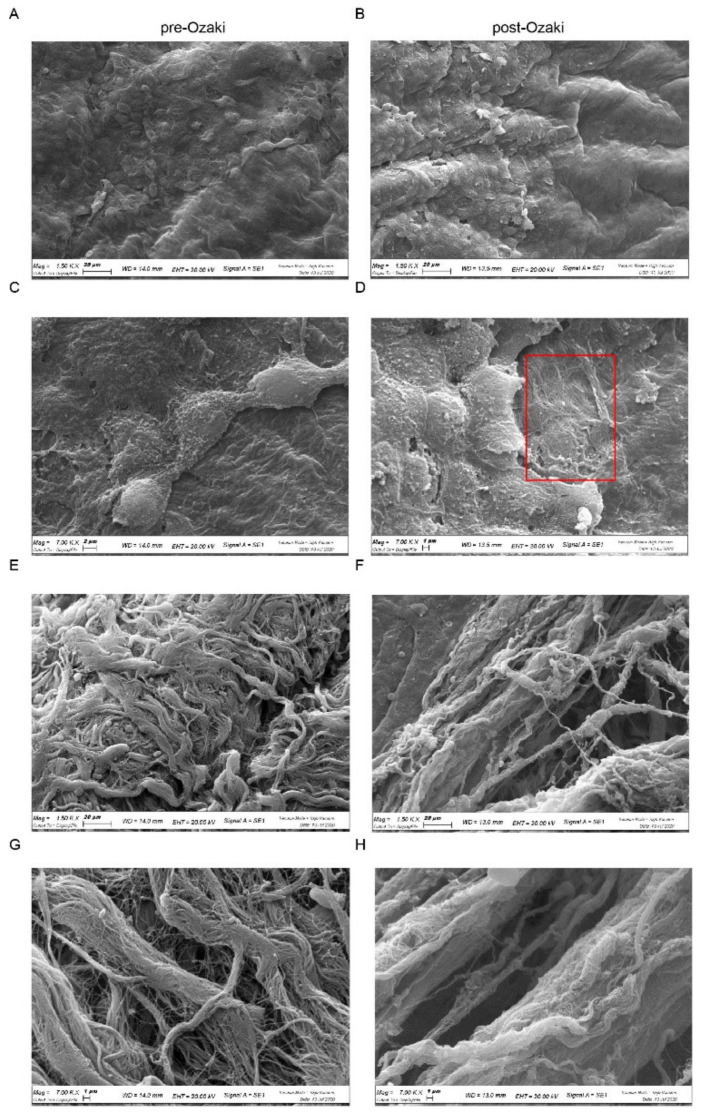
SEM morphological analysis of pre-Ozaki and post-Ozaki samples. (**A**,**C**) Representative images of the serosa layer in the pre-Ozaki pericardium show a continuous layer of mesothelial cells, whose surface is covered by short and abundant microvilli. (**B**,**D**) This layer appears interrupted in some areas (red box) in the post-Ozaki pericardium. (**E**,**G**) Representative images of the surface of the fibrosa layer in the pre-Ozaki pericardium display overlapping wavy collagen and elastic fibers. (**F**,**H**) These fibers assume a more parallel and compact arrangement in the post-Ozaki pericardium. Representative images at 1500× (**A**,**B**,**E**,**F**) and 7000× (**C**,**D**,**G**,**H**) magnifications.

**Figure 3 jcm-10-03954-f003:**
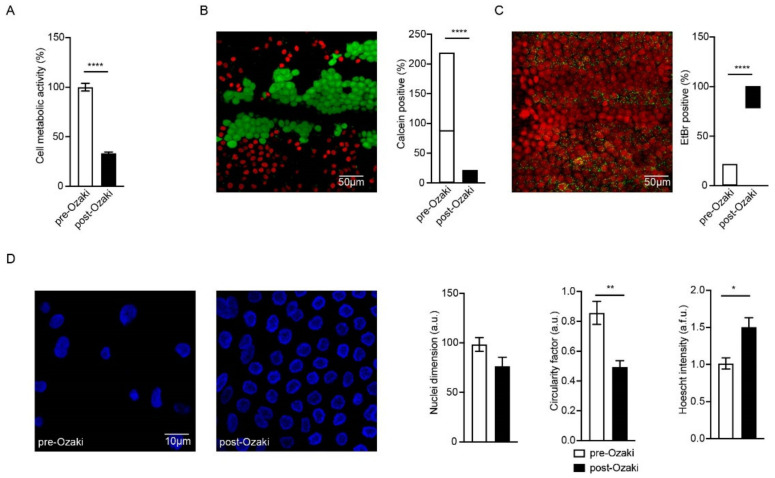
Evaluation of cell viability in peri cardial samples after treatment with GA (post-Ozaki) in comparison to native pericardium (pre-Ozaki). (**A**) MTT assay shows that residual cell viability in post-Ozaki samples is 33.32 ± 1.29% when compared to that of pre-Ozaki samples. **** *p* < 0.0001; *n* = 15. (**B**,**C**) Representative images of pre-Ozaki and post-Ozaki samples, respectively, showing live (green) and dead (red) cells, and their quantification (graphs in panel (**B**) and panel (**C**), respectively). **** *p* < 0.0001; *n* = 15. (**D**) Representative images of nuclear morphology following Hoechst staining (blue) and evaluation of nuclei dimension (left graph), circularity factor (center graph), and Hoechst intensity (right graph). * *p* < 0.05, ** *p* < 0.01; *n* = 15.

**Figure 4 jcm-10-03954-f004:**
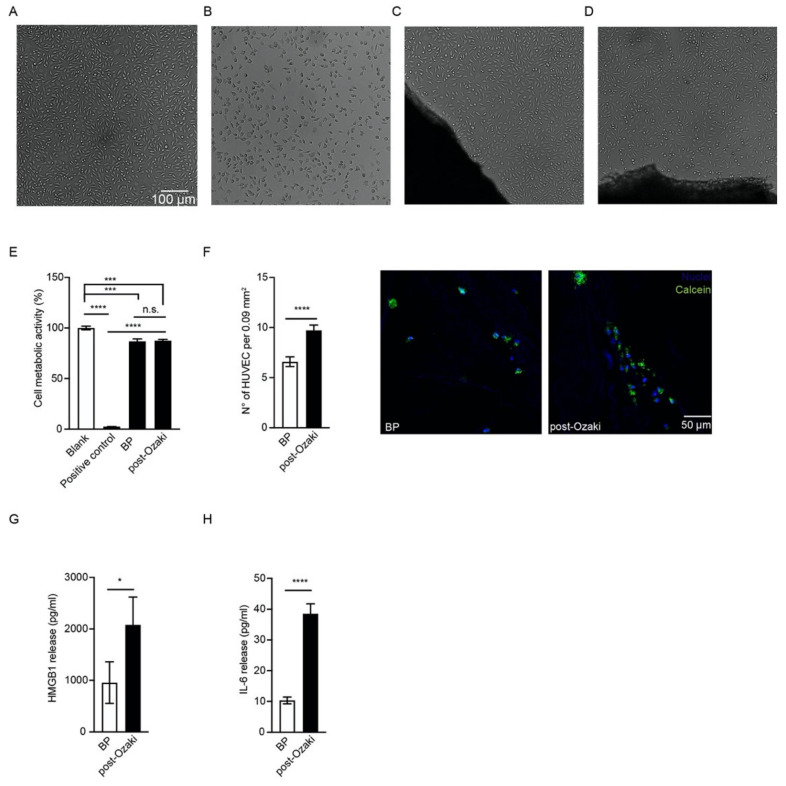
Citotoxicity assay, endothelialization, and expression of pro-inflammatory mediators in the post-Ozaki pericardium and BP control membranes. Morphology of L929 mouse fibroblasts cultured (**A**) without tested materials (blank), (**B**) in the presence of 0.6% GA (positive control), or grown in contact with (**C**) the commercial BP membrane (negative control) and (**D**) post-Ozaki samples. (**E**) Quantification of cytotoxicity by the MTT assay. Data are expressed as a percentage of viability referred to control cells growing without tested materials. *** *p* < 0.001, **** *p* < 0.0001; *n* = 15. (**F**) Quantification of endothelial cell number on the fibrous side of pericardium 4 days after seeding. **** *p* < 0.0001; *n* = 15. Representative images of HUVEC, labelled with the vital dye calcein-AM (in green) and the nuclear dye Hoechst (in blue), adhering on the BP membrane (left) and on post-Ozaki pericardium (right). Quantification of (**G**) HMGB1 and (**H**) IL-6 from cell culture supernatant after 4 days from seeding. * *p* < 0.05, **** *p* < 0.0001; *n* = 15.

## Data Availability

Data is contained within the article.

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
