# Peer review of "Biological Characterization of Human Autologous Pericardium Treated with the Ozaki Procedure for Aortic Valve Reconstruction"

_jcm, 2021, doi:10.3390/jcm10173954_

Round 1

Reviewer 1 Report

Thank you for allowing me to review this elegant manuscript entitled: Biological characterization of human autologous pericardium treated with the Ozaki procedure for aortic valve reconstruction.

- Have you checked normality of the data? Which did you use? If the data are not normally distributed, you need to express data with median with IQR and need to compare the data with appropriate test, not student T test.

-Otherwise, this manuscript is well written, I don’t have concerns.

Author Response

We thank the reviewers for their constructive comments, which helped to improve the quality, structure and content of the manuscript.

Reviewers’suggestions are highlighted in red through the text.

Ref 1

Thank you for allowing me to review this elegant manuscript entitled: Biological characterization of human autologous pericardium treated with the Ozaki procedure for aortic valve reconstruction.

- Have you checked normality of the data? Which did you use? If the data are not normally distributed, you need to express data with median with IQR and need to compare the data with appropriate test, not student T test.

The reviewer is right. We truly agree with him/her. To address this statistical issue, we have checked the normality of the data using the Shapiro-Wilk normality test (see “2.9. Statistical analysis” paragraph). We detected non-normal distribution only in data showed in Figure 3B and 3C (i.e. live/dead assay). In this case, median with IQR has been represented in the new figure, and values have been reported in the text. Statistical significance has been calculated by the Mann-Whitney U test.

-Otherwise, this manuscript is well written, I don’t have concerns.

Reviewer 2 Report

I read with high interest the work "Biological characterization of human autologous pericardium treated with the Ozaki procedure for aortic valve reconstruction" by Chiara Gardin et. al..

The authors seek to illuminate the effect of gultaraldehyde (GA) when used to prep autologous pericardium in the setting of aortic valve reconstruction, the so called Ozaki procedure. 

They use a set of seven experiments to gain different angles to view the issue, examining structure (2.2. histology, 2.3 electro microscopy), cell viability (2.4. MTT, 2.5. cell death, 2.6. properties of the cell nucleus), and effetcs on surrounding tissues (2.7 cytotoxicity, 2.8. HUVEC colonization).

The paper is very well written and also picks up readers who are not proficient in the world of bench-work and basic / translational science (like myself).

There are a few comments on structure and content, and also to even further understandability.

  • from line 71, I suggest to move the whole paragraph starting with " Native pericardium was used as..." to the section "2. Materials and Methods"
  • Alike, I suggest to move the paragraph from line 298 "Nonetheless, it is also true that a significant..." to the section "4. Discussion"
  • Please check if SEM is a common abbreviation for "scanning electron microscopy". Otherwise, this should stay reserved for the statistical term "standard error of the means". Maybe "EM" for electron microscopy would suffice?
  • The study involves using live human tissue for researach. I think, apart from the general patient consent, an approval vote of the local ethics committee would be advisable
  • The definition of the different groups is somewhat redundant. Along with this, the introducing paragraph from line 86, starting with "Human pericardium samples (7x8 cm) were separated from surrounding tissues, resected and fixed onto plates...." is somewhat hard to understand. From my understanding, the area of 7 x 8 cm pericardium seems to be the standard needed for the procedure (compare also 10.5606/tgkdc.dergisi.2019.01903, incl. video). This is obviously needed for three things in the study: 1) the pre-Ozaki samples (2 x 2 cm), 2) the post-Ozaki samples (whatever is left after 3)), and 3) material to manufacture the neo-leaflets for the reconstruction. I suggest rephrasing. For instance, that first the pericardium was harvested in the typical fashion, then a native 2 x 2 sample was removed and stored for later comparison (pre-Ozaki samples), then the remaining tissue was treated with GA according to the literature [specifics]. The tissue then was used to create the neo-leaflets, with the rest of the material composing the post-Ozaki samples. If I reflected the process correctly, you could omit the repetition in the 3. Results section "hereafter named as...").
  • In 2.3. SEM, I do not understand why the tissues (or a seperate tissue, for that matter) needed to be fixed with 2 % GA preped in 0.1 M HEPES. I would expect the use of the priously prepared pre- and post-Ozaki examples. If the use of GA is a prerequisite for the use of SEM ("fixation"), then would the use of GA, the same agent studied in this experiment, not interfere with the results, esp. in the pre-Ozaki samples group? Please discuss if the use of GA is a prerequisite for SEM.
  • At some points, text and figures, the effect of GA on the alignment of the collagen is referred to as "(more) aligned" or "clearer". I think, what you might want to say is, that GA causes a (more) parallel alignment of the collagen, thus leading to compactation of the ECM (as opposed to "staggered", "scaled", "criss-crossed", or "transversed" alignment without GA), n'est pas? 

Figures

In general, recheck figures and corresponding text. In my opinion, at the moment the figure text also needs the manuscript text in order to give the whole information. However, the figure text should be able to stand and explain by itself.

  • Fig. 1: It seems obvious that there is no loss in tissue, i.e. ECM, comparing pre- to post-Ozaki samples. Therefore, I think the word "thinner" might be misleading. Perhaps "smaller diameter" or similar would be more neutral. Also, as later pointed out in the manuscript text, the images provided already suggest rather denser, compacted tissue. So I suggest to mention this in the figure. Please also allude to the disrupted serosa, nicely depicted in Panel B.
  • Fig. 2 is very nice, please refer to the alignment issue above
  • Fig. 3 is also very informative but at this stage overburdened. I suggest to at least use Panels A-D, with C given to the red image, showing the EtBr positive (post-Ozaki?) samples, and D what is now C. This brings me to Panel B, as it is now: it is unclear which sample is used, as the figure text refers to both, pre- and post-Ozaki samples. Please be specific. You might consider using a Panel E for the graphs on the lower right. I understand however, that they belong to the images to the left of now C. So, at least, when referring to, please address specifically, for instance as left, right, center "graph". Also, as mentioned above, please recheck the manuscript text also, so it clearly guides through the figure.
  • Please rephrase the paragraph beginning from 273 "Post-Ozaki samplex, as well as the commercial ...", so that the reference to the figure is in the correct order, i.e. 4C and D, respectively (as the order in the figure makes the most sense). 
  • The growth of HUVEC cells on the tissues probably cannot be seen synonymous with (gerneral) "cellular repopulation". One might rather think that  it represents endothelialisation. To assume that cells died from GA use, as shown in the experiments on cell viability and cell death, are replaced later on, because HUVECs grow on the tissue, is speculative, at best.
  • A thought for the discussion: you note that GA is routinely used for the preparation of other prostheses, including AVR protheses. You might want to use the opportunity to speculate on the durability of the AVreconstruction vs. AVReplacement, as your findings suggest some advantages for the Ozaki procedure. 

Author Response

I read with high interest the work "Biological characterization of human autologous pericardium treated with the Ozaki procedure for aortic valve reconstruction" by Chiara Gardin et. al..

The authors seek to illuminate the effect of gultaraldehyde (GA) when used to prep autologous pericardium in the setting of aortic valve reconstruction, the so called Ozaki procedure.

They use a set of seven experiments to gain different angles to view the issue, examining structure (2.2. histology, 2.3 electro microscopy), cell viability (2.4. MTT, 2.5. cell death, 2.6. properties of the cell nucleus), and effetcs on surrounding tissues (2.7 cytotoxicity, 2.8. HUVEC colonization).

The paper is very well written and also picks up readers who are not proficient in the world of bench-work and basic / translational science (like myself).

There are a few comments on structure and content, and also to even further understandability.

  • from line 71, I suggest to move the whole paragraph starting with " Native pericardium was used as..." to the section "2. Materials and Methods"

The sentence has been moved to the section "2. Materials and Methods".

  • Alike, I suggest to move the paragraph from line 298 "Nonetheless, it is also true that a significant..." to the section "4. Discussion".

The paragraph has been moved to the section "4. Discussion".

  • Please check if SEM is a common abbreviation for "scanning electron microscopy". Otherwise, this should stay reserved for the statistical term "standard error of the means". Maybe "EM" for electron microscopy would suffice?

SEM is a common abbreviation for "scanning electron microscopy" and it is necessary to distinguish it from TEM, which is the acronym for "transmission electron microscopy". Consequently, the statistical term "standard error of the mean" has been abbreviated as SE.

  • The study involves using live human tissue for researach. I think, apart from the general patient consent, an approval vote of the local ethics committee would be advisable.

In the section “2. Materials and Methods", paragraph “2.1. Sample collection and processing”, we have specified that the study has been approved by the local ethics committee, and we have provided both the protocol code and date of approval. This information is also present in the “Institutional Review Board Statement” at the end of the manuscript.

  • The definition of the different groups is somewhat redundant. Along with this, the introducing paragraph from line 86, starting with "Human pericardium samples (7x8 cm) were separated from surrounding tissues, resected and fixed onto plates...." is somewhat hard to understand. From my understanding, the area of 7 x 8 cm pericardium seems to be the standard needed for the procedure (compare also 10.5606/tgkdc.dergisi.2019.01903, incl. video). This is obviously needed for three things in the study: 1) the pre-Ozaki samples (2 x 2 cm), 2) the post-Ozaki samples (whatever is left after 3)), and 3) material to manufacture the neo-leaflets for the reconstruction. I suggest rephrasing. For instance, that first the pericardium was harvested in the typical fashion, then a native 2 x 2 sample was removed and stored for later comparison (pre-Ozaki samples), then the remaining tissue was treated with GA according to the literature [specifics]. The tissue then was used to create the neo-leaflets, with the rest of the material composing the post-Ozaki samples. If I reflected the process correctly, you could omit the repetition in the 3. Results section "hereafter named as...").

The reviewer interpreted correctly the procedure; therefore, based on his/her suggestions, corrections to this consideration have been made in the text.

  • In 2.3. SEM, I do not understand why the tissues (or a seperate tissue, for that matter) needed to be fixed with 2 % GA preped in 0.1 M HEPES. I would expect the use of the priously prepared pre- and post-Ozaki examples. If the use of GA is a prerequisite for the use of SEM ("fixation"), then would the use of GA, the same agent studied in this experiment, not interfere with the results, esp. in the pre-Ozaki samples group? Please discuss if the use of GA is a prerequisite for SEM.

The first step in preparing a biological sample for SEM requires its fixation with a fixing agent. Buffered aldehyde, typically 2%-5% GA, is the chemical fixative most still used today [https://doi.org/10.1016/0739-6260(86)90042-0]. This preliminary step is necessary to keep both surface and intracellular structural details intact before further processing, and it is particularly important for native specimens, such as the pre-Ozaki pericardium. For a better comparison between the samples, we treated post-Ozaki tissues in the same way. Several published papers employed such a treatment for tissues previously fixed with GA [PMID: 33231114; PMID: 30392024; PMID: 29273368]. Other fixatives used for SEM include paraformaldehyde and osmium tetroxide. Nevertheless, paraformaldehyde is generally used in combination with GA because it penetrates tissues rapidly but only mildly stabilizes proteins, which are instead permanently fixed by GA. On the other hand, osmium tetroxide is most often used as a secondary fixative after GA, but it is more dangerous than GA and can induce some surface damage due to its strong oxidizing properties.

  • At some points, text and figures, the effect of GA on the alignment of the collagen is referred to as "(more) aligned" or "clearer". I think, what you might want to say is, that GA causes a (more) parallel alignment of the collagen, thus leading to compactation of the ECM (as opposed to "staggered", "scaled", "criss-crossed", or "transversed" alignment without GA), n'est pas?

The reviewer’s interpretation is correct. Consequently, we have modified the term “aligned” in “a more parallel arrangement” or in “a more compact alignment” of the ECM in the text.

Figures

In general, recheck figures and corresponding text. In my opinion, at the moment the figure text also needs the manuscript text in order to give the whole information. However, the figure text should be able to stand and explain by itself.

  • 1: It seems obvious that there is no loss in tissue, i.e. ECM, comparing pre- to post-Ozaki samples. Therefore, I think the word "thinner" might be misleading. Perhaps "smaller diameter" or similar would be more neutral. Also, as later pointed out in the manuscript text, the images provided already suggest rather denser, compacted tissue. So I suggest to mention this in the figure. Please also allude to the disrupted serosa, nicely depicted in Panel B.

In the text of Fig. 1, we have detailed the effect of GA on the serosa and fibrosa layers of post-Ozaki pericardium, as suggested by the reviewer.

  • 2 is very nice, please refer to the alignment issue above.

In the text of Fig. 2, we have emphasized the alignment of collagen and elastic fibers caused by the Ozaki procedure.

  • 3 is also very informative but at this stage overburdened. I suggest to at least use Panels A-D, with C given to the red image, showing the EtBr positive (post-Ozaki?) samples, and D what is now C. This brings me to Panel B, as it is now: it is unclear which sample is used, as the figure text refers to both, pre- and post-Ozaki samples. Please be specific. You might consider using a Panel E for the graphs on the lower right. I understand however, that they belong to the images to the left of now C. So, at least, when referring to, please address specifically, for instance as left, right, center "graph". Also, as mentioned above, please recheck the manuscript text also, so it clearly guides through the figure.

We have appreciated the reviewer’s suggestions. Therefore, we have specified that image of Panel B represents pre-Ozaki whereas the image of the new Panel C represents post-Ozaki, both stained with live/dead staining solutions. Consequently, the old Panel C has become Panel D. Both the manuscript and text of Fig. 3 have been modified accordingly.

  • Please rephrase the paragraph beginning from 273 "Post-Ozaki samplex, as well as the commercial ...", so that the reference to the figure is in the correct order, i.e. 4C and D, respectively (as the order in the figure makes the most sense).

The paragraph has been rephrased.

  • The growth of HUVEC cells on the tissues probably cannot be seen synonymous with (gerneral) "cellular repopulation". One might rather think that it represents endothelialisation. To assume that cells died from GA use, as shown in the experiments on cell viability and cell death, are replaced later on, because HUVECs grow on the tissue, is speculative, at best.

We agree with the reviewer that the term “endothelialization” is more appropriate that “cellular repopulation” in this context. Changes have been made in the text.

  • A thought for the discussion: you note that GA is routinely used for the preparation of other prostheses, including AVR protheses. You might want to use the opportunity to speculate on the durability of the AVreconstruction vs. AVReplacement, as your findings suggest some advantages for the Ozaki procedure.

We thank the reviewer for this consideration. We believe that the reconstruction of aortic valve with the Ozaki procedure represents an attractive option when compared to aortic valve replacement because there is no need to use foreign material (as for biological prostheses) or post-operative anticoagulant therapy (as for mechanical prostheses). In addition, and as stated in the section “1. Introduction”, the strength of this technique relies on the independent replacement of each cusp, thus better reflecting the native valvular anatomy and motion. Despite these premises and the encouraging results obtained in this study, we are aware that further in vitro experiments and long-term follow-up studies involving a large cohort are required to prove the durability of aortic valves reconstructed with the Ozaki procedure.